# Fast Inference and Transfer of Compositional Task Structures for Few-shot Task Generalization

**Sungryull Sohn**[1,2]   **Hyunjae Woo**[1]   **Jongwook Choi**[1]   **Lyubing Qiang**[1]   **Izzeddin Gur**[3]   **Aleksandra Faust**[3]

**Honglak Lee**[1,2]

[1]University of Michigan
[2]LG AI Research Center Ann Arbor
[3]Google Research

## Abstract

We tackle real-world problems with complex structures beyond the pixel-based game or simulator. We formulate it as a few-shot reinforcement learning problem where a task is characterized by a subtask graph that defines a set of subtasks and their dependencies that are unknown to the agent. Different from the previous meta-RL methods trying to directly infer the unstructured task embedding, our multi-task subtask graph inferencer (MTSGI) first infers the common high-level task structure in terms of the subtask graph from the training tasks, and use it as a prior to improve the task inference in testing. Our experiment results on 2D grid-world and complex web navigation domains show that the proposed method can learn and leverage the common underlying structure of the tasks for faster adaptation to the unseen tasks than various existing algorithms such as meta reinforcement learning, hierarchical reinforcement learning, and other heuristic agents.

## 1 INTRODUCTION

Recently, deep reinforcement learning (RL) has shown an outstanding performance on various domains such as video games [Mnih et al., 2015, Vinyals et al., 2019] and board games [Silver et al., 2017]. However, most of the successes of deep RL were focused on a single-task setting where the agent is allowed to interact with the environment for hundreds of millions of time steps. In numerous real-world scenarios, interacting with the environment is expensive or limited, and the agent is often presented with a novel task that is not seen during its training time. To overcome this limitation, many recent works focused on scaling the RL algorithm beyond the single-task setting. Recent works on multi-task RL aim to build a single, contextual policy that can solve multiple related tasks and generalize to unseen tasks. However, they require a certain form of task embedding as an extra input that often fully characterizes the given task [Oh et al., 2017, Andreas et al., 2017, Yu et al., 2017, Chaplot et al., 2018], or requires a human demonstration Huang et al. [2018], which are not readily available in practice. Meta RL [Finn et al., 2017, Duan et al., 2016] focuses on a more general setting where the agent should learn about the unseen task purely via interacting with the environment without any additional information. However, such meta-RL algorithms either require a large amount of experience on the diverse set of tasks or are limited to a relatively smaller set of simple tasks with a simple task structure.

On the contrary, real-world problems require the agent to solve much more complex and compositional tasks without human supervision. Consider a web-navigating RL agent given the task of checking out the products from an online store as shown in Figure 1. The agent can complete the task by filling out the required web elements with the correct information such as shipping or payment information, navigating between the web pages, and placing the order. Note that the task consists of multiple *subtasks* and the subtasks have complex dependencies in the form of *precondition*; for instance, the agent may proceed to the payment web page (see *Bottom*, B) *after* all the required shipping information has been correctly filled in (see *Bottom*, A), or the `credit_card_number` field will appear *after* selecting the `credit_card` as a payment method (see *Top, Middle* in Figure 1). Learning to perform such a task can be quite challenging if the reward is given only after yielding meaningful outcomes (*i.e.*, sparse reward task). This is the problem scope we focus on in this work: solving and generalizing to unseen compositional sparse-reward tasks with complex subtask dependencies without human supervision.

Recent works [Sohn et al., 2019, Xu et al., 2017, Huang et al., 2018, Liu et al., 2016, Ghazanfari and Taylor, 2017] tackled the compositional tasks by explicitly inferring the underlying task structure in a graph form. Specifically, the

*Accepted for the 38th Conference on Uncertainty in Artificial Intelligence* (UAI 2022).

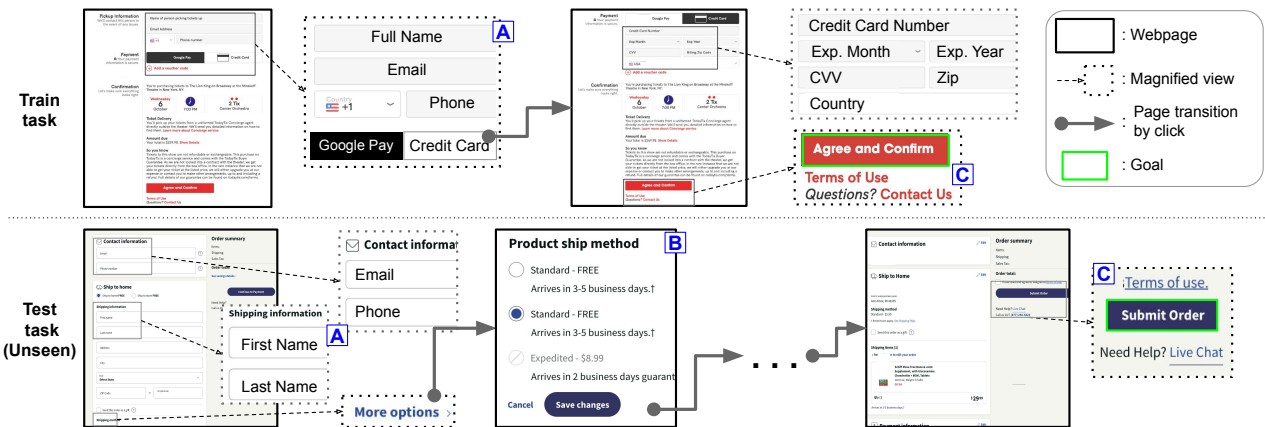

Figure 1: An illustration of the train *(Top)* and test task *(Bottom)* in our *SymWoB* domain. Some selected actionable web-elements (*e.g.*, text fields and buttons) are magnified (dotted arrow and box) for readability. The agent's goal (green box) is to checkout the products in unseen test website by interacting with the web elements in a correct order. For example, in train task, the agent should fill out all the text fields in (*Top*, **A**) **before** clicking the credit_card button to transition (gray arrow) to next page. The high-level checkout processes in different websites have many commonalities while certain details may differ. For example, in both train and test tasks, the agent should fill out the user information (*Top* and *Bottom*, **A**) before proceeding to the next page or there exist similar elements (*Top* and *Bottom*, **C**). However, the details may differ; *e.g.*, the train task (*Top*, **A**) has a single text field for full name, while the test task (*Bottom*, **A**) has separate text fields for the first and last name, respectively. Also, only the test website (*Bottom*, **B**) requires shipping information since the training website does not ship the product.

subtask graph inference (SGI) framework [Sohn et al., 2019] uses inductive logic programming (ILP) on the agent's own experience to infer the task structure in terms of *subtask graph* and learns a contextual policy to *execute* the inferred task in few-shot RL setting. However, it only meta-learned the adaptation policy that relates to the efficient exploration, while the task inference and execution policy learning were limited to a single task (*i.e.*, both task inference and policy learning were done from scratch for each task), limiting its capability of handling large variance in the task structure. We claim that the inefficient task inference may hinder applying the SGI framework to a more complex domain such as web navigation [Shi et al., 2017, Liu et al., 2018] where a task may have a large number of subtasks and complex dependencies between them. We note that humans can navigate an unseen website by transferring the high-level process learned from previously seen websites.

Inspired by this, we extend the SGI framework to a *multi-task subtask graph inferencer* (MTSGI) that can generalize the previously learned task structure to the unseen task for faster adaptation and stronger generalization. Figure 2 outlines our method. MTSGI estimates the prior model of the subtask graphs from the training tasks. When an unseen task is presented, MTSGI samples the prior that best matches with the current task, and incorporates the sampled prior model to improve the latent subtask graph inference, which in turn improves the performance of the evaluation policy. We demonstrate results in the 2D grid-world domain and the web navigation domain that simulates the interaction with 15

actual websites. We compare our method with MSGI [Sohn et al., 2019] that learns the task hierarchy from scratch for each task, and two other baselines including hierarchical RL and a heuristic algorithm. We find that MTSGI significantly outperforms all other baselines, and the learned prior model enables more efficient task inference compared to MSGI.

## 2 PRELIMINARIES

**Few-shot Reinforcement Learning** A *task* is defined by an MDP $\mathcal{M}_G = (\mathcal{S}, \mathcal{A}, \mathcal{P}_G, \mathcal{R}_G)$ parameterized by a task parameter $G$ with a set of states $\mathcal{S}$, a set of actions $\mathcal{A}$, transition dynamics $\mathcal{P}_G$, reward function $\mathcal{R}_G$. The goal of $K$-shot RL [Duan et al., 2016, Finn et al., 2017], is to efficiently solve a distribution of unseen test tasks $\mathcal{M}^{\text{test}}$ by learning and transferring the common knowledge from the training tasks $\mathcal{M}^{\text{train}}$. It is assumed that the training and test tasks do not overlap (*i.e.*, $\mathcal{M}^{\text{train}} \cap \mathcal{M}^{\text{test}} = \emptyset$) but share a certain commonality such that the knowledge learned from the training tasks may be helpful for learning the test tasks. For each task $\mathcal{M}_G$, the agent is given $K$ steps budget for interacting with the environment. During meta-training, the goal of multi-task RL agent is to learn a prior (*i.e.*, slow-learning) over the training tasks $\mathcal{M}^{\text{train}}$. Then, the learned prior may be exploited during the meta-test to enable faster adaptation on unseen test tasks $\mathcal{M}^{\text{test}}$. For each task, the agent faces two phases: an *adaptation phase* where the agent learns a task-specific behavior (*i.e.*, fast-learning) for $K$ environment steps, which often spans over multiple episodes, and a

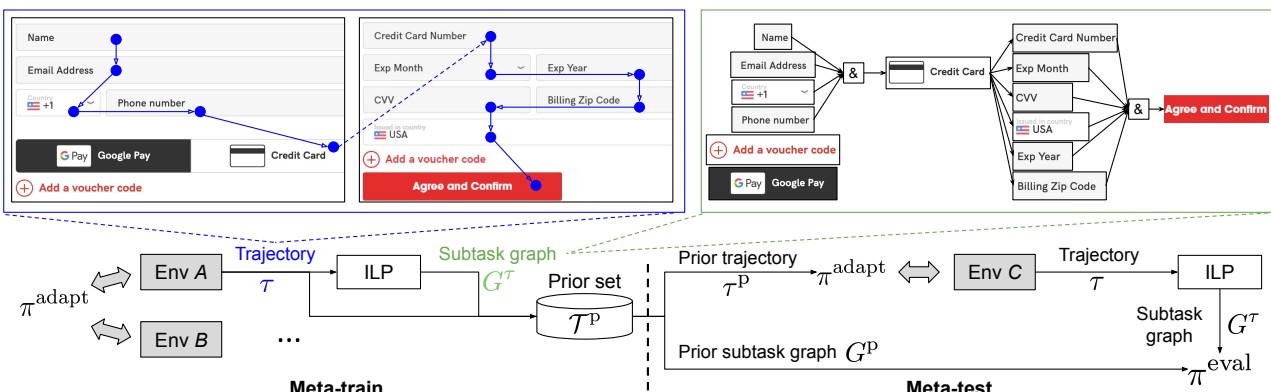

Figure 2: The overview of our algorithm and the example of agent's trajectory and the inferred subtask graph. In meta-train *(Left)*, the adaptation policy $\pi^{\text{adapt}}$ interacts with the environment and collects the trajectory $\tau$. The inductive logic programming (ILP) module takes as input the trajectory, and infers the task structure in terms of the subtask graph $G^\tau$. The trajectory and the subtask graph are stored as a prior. In meta-testing *(Right)*, the adaptation policy incorporates the prior trajectory $\tau^{\text{p}}$ to efficiently explore the environment, and ILP module infers the subtask graph $G^\tau$ from the adaptation trajectory $\tau$. Finally, the evaluation policy $\pi^{\text{eval}}$ takes as input the prior and inferred subtask graphs $(G^{\text{p}}, G^\tau)$ to solve the test task.

*evaluation phase* where the adapted behavior is evaluated. In the evaluation phase, the agent is not allowed to perform any form of learning, and agent's performance on the task $\mathcal{M}_G$ is measured in terms of the return:

$$\mathcal{R}_{\mathcal{M}_G}(\pi_{\phi_K}) = \mathbb{E}_{\pi_{\phi_K}, \mathcal{M}_G}\left[\sum_{t=1}^{H} r_t\right], \quad (1)$$

where $\pi_{\phi_K}$ is the policy after $K$ update steps of adaptation, $H$ is the horizon of evaluation phase, and $r_t$ is the reward at time $t$ in the evaluation phase.

## 3 SUBTASK GRAPH INFERENCE PROBLEM

The *subtask graph inference* problem [Sohn et al., 2019] is a few-shot RL problem where a task is parameterized by a set of subtasks and their dependencies. Formally, a task consists of $N$ subtasks $\mathbf{\Phi} = \{\Phi^1, \ldots, \Phi^N\}$, and each subtask $\Phi^i$ is parameterized by a tuple $(\mathcal{S}_{\text{comp}}^i, G_{\mathbf{c}}^i, G_{\mathbf{r}}^i)$. The *goal state* $\mathcal{S}_{\text{comp}}^i \subset \mathcal{S}$ and *precondition* $G_{\mathbf{c}}^i : \mathcal{S} \rightarrow \{0, 1\}$ defines the condition that a subtask is *completed*: the current state should be contained in its goal states (*i.e.*, $\mathbf{s}_t \in \mathcal{S}_{\text{comp}}^i$) and the precondition should be satisfied (*i.e.*, $G_{\mathbf{c}}^i(\mathbf{s}_t) = 1$). If the precondition is not satisfied (*i.e.*, $G_{\mathbf{c}}^i(\mathbf{s}_t) = 0$), the subtask cannot be completed and the agent receives no reward even if the goal state is achieved. The *subtask reward function* $G_{\mathbf{r}}^i$ defines the amount of reward given to the agent when it *completes* the subtask $i$: $r_t \sim G_{\mathbf{r}}^i$. We note that the subtasks $\{\Phi^1, \ldots, \Phi^N\}$ are unknown to the agent. Thus, the agent should learn to infer the underlying task structure and complete the subtasks in an optimal order while satisfying the required preconditions.

**State** In the subtask graph inference problem, it is assumed that the state input provides the high-level status of

the subtasks. Specifically, the state consists of the followings: $\mathbf{s}_t = (\text{obs}_t, \mathbf{x}_t, \mathbf{e}_t, \text{step}_{\text{epi},t}, \text{step}_{\text{phase},t})$. The $\text{obs}_t \in \{0, 1\}^{W \times H \times C}$ is a visual observation of the environment. The completion vector $\mathbf{x}_t \in \{0, 1\}^N$ indicates whether each subtask is complete. The eligibility vector $\mathbf{e}_t \in \{0, 1\}^N$ indicates whether each subtask is eligible (*i.e.*, precondition is satisfied). Following the few-shot RL setting, the agent observes two scalar-valued time features: the remaining time steps until the episode termination $\text{step}_{\text{epi},t} \in \mathbb{R}$ and the remaining time steps until the phase termination $\text{step}_{\text{phase},t} \in \mathbb{R}$.

**Options** For each subtask $\Phi^i$, the agent can learn an option $\mathcal{O}^i$ [Sutton et al., 1999] that reaches the goal state of the subtask. Following Sohn et al. [2019], such options are pre-learned individually by maximizing the goal-reaching reward: $r_t = \mathbb{I}(\mathbf{s}_t \in \mathcal{S}_{\text{comp}}^i)$. At time step $t$, we denote the option taken by the agent as $\mathbf{o}_t$ and the binary variable that indicates whether episode is terminated as $d_t$.

## 4 METHOD

We propose a novel Multi-Task Subtask Graph Inference (MTSGI) framework that can perform an efficient inference of latent task embedding (*i.e.*, subtask graph). The overall method is outlined in Figure 2. Specifically, in meta-training, MTSGI models the prior in terms of (1) adaptation trajectory $\tau$ and (2) subtask graph $G$ from the agent's experience. In meta-testing, MTSGI samples (1) the prior trajectory $\tau^{\text{p}}$ for more efficient exploration in adaptation and (2) the prior subtask graph $G^{\text{p}}$ for more accurate task inference.

**Algorithm 1** Meta-training: learning the prior

**Require:** Adaptation policy $\pi^{\text{adapt}}$
**Ensure:** Prior set $\mathcal{T}^{\text{p}}$
1: $\mathcal{T}^{\text{p}} \leftarrow \emptyset$
2: **for** each task $\mathcal{M} \in \mathcal{M}^{\text{train}}$ **do**
3:     Rollout adaptation policy:
    $\tau = \{\mathbf{s}_t, \mathbf{o}_t, r_t, d_t\}_{t=1}^K \sim \pi^{\text{adapt}}$ in task $\mathcal{M}$
4:     Infer subtask graph $G^\tau = \arg\max_G p(\tau|G)$
5:     $\pi^{\text{eval}} = \text{GRProp}(G^\tau)$
6:     Evaluate the agent: $\tau^{\text{eval}} \sim \pi^{\text{eval}}$ in task $\mathcal{M}$
7:     Update prior $\mathcal{T}^{\text{p}} \leftarrow \mathcal{T}^{\text{p}} \cup (G^\tau, \tau)$
8: **end for**

**Algorithm 2** meta-testing: multi-task SGI

**Require:** Adaptation policy $\pi^{\text{adapt}}$, prior set $\mathcal{T}^{\text{p}}$
1: **for** each task $\mathcal{M} \in \mathcal{M}^{\text{test}}$ **do**
2:     Sample prior: $(G^{\text{p}}, \tau^{\text{p}}) \sim p(\mathcal{T}^{\text{p}})$
3:     Rollout adaptation policy:
    $\tau = \{\mathbf{s}_t, \mathbf{o}_t, r_t, d_t\}_{t=1}^K \sim \pi^{\text{adapt}}$ in task $\mathcal{M}$
4:     Infer subtask graph $G^\tau = \arg\max_G p(\tau|G)$
5:     $\pi^{\text{eval}}(\cdot|\tau, \tau^{\text{p}}) \propto \text{GRProp}(\cdot|G^\tau)^\alpha \text{GRProp}(\cdot|G^{\text{p}})^{(1-\alpha)}$
6:     Evaluate the agent: $\tau^{\text{eval}} \sim \pi^{\text{eval}}$ in task $\mathcal{M}$
7: **end for**

## 4.1 MULTI-TASK ADAPTATION POLICY

The goal of *adaptation policy* is to efficiently explore and gather the information about the task. Intuitively, if the adaptation policy completes more diverse subtasks, then it can provide more data to the task inference module (ILP), which in turn can more accurately infer the task structure. To this end, we extend the upper confidence bound (UCB)-based adaptation policy proposed in Sohn et al. [2019] as follows:

$$\pi^{\text{adapt}}(o = \mathcal{O}^i \mid s) \propto \exp\left(r^i + \sqrt{2}\frac{\log\left(\sum_j n^j\right)}{n^i}\right), \tag{2}$$

where $r^i$ is the empirical mean of the reward received after executing subtask $i$ and $n^i$ is the number of times subtask $i$ has been executed within the current task. Note that the exploration parameters $\{r^i, n^i\}_{i=1}^N$ can be computed from the agent's trajectory. In meta-train, the exploration parameters are initialized to zero when a new task is sampled. In meta-test, the exploration parameters are initialized with those of the sampled prior. Intuitively, this helps the agent visit novel states that were unseen during meta-training.

## 4.2 META-TRAIN: LEARNING THE PRIOR SUBTASK GRAPH

Let $\tau$ be an adaptation trajectory of the agent for $K$ steps. The goal is to infer the latent subtask graph $G$ for the given training task $\mathcal{M}_G \in \mathcal{M}^{\text{train}}$, specified by preconditions $G_{\mathbf{c}}$ and subtask rewards $G_{\mathbf{r}}$. We find the maximum-likelihood estimate (MLE) of $G = (G_{\mathbf{c}}, G_{\mathbf{r}})$ that maximizes the likelihood of the adaptation trajectory $\tau$:

$$\widehat{G}^{\text{MLE}} = \arg\max_{G_{\mathbf{c}}, G_{\mathbf{r}}} p(\tau|G_{\mathbf{c}}, G_{\mathbf{r}}). \tag{3}$$

Following Sohn et al. [2019], we infer the precondition $G_{\mathbf{c}}$ and the subtask reward $G_{\mathbf{r}}$ as follows (See Appendix for the

detailed derivation):

$$\widehat{G}_{\mathbf{c}}^{\text{MLE}} = \arg\max_{G_{\mathbf{c}}} \prod_{t=1}^H p(\mathbf{e}_t|\mathbf{x}_t, G_{\mathbf{c}}), \tag{4}$$

$$\widehat{G}_{\mathbf{r}}^{\text{MLE}} = \arg\max_{G_{\mathbf{r}}} \prod_{t=1}^H p(r_t|\mathbf{e}_t, \mathbf{o}_t, G_{\mathbf{r}}). \tag{5}$$

where $\mathbf{e}_t$ is the eligibility vector, $\mathbf{x}_t$ is the completion vector, $\mathbf{o}_t$ is the option taken, $r_t$ is the reward at time step $t$.

**Precondition inference** The problem in Equation (4) is known as the inductive logic programming (ILP) problem that finds a boolean function that satisfies all the indicator functions. Specifically, $\{\mathbf{x}_t\}_{t=1}^H$ forms binary vector inputs to programs, and $\{e_t^i\}_{t=1}^H$ forms Boolean-valued outputs of the $i$-th program that predicts the eligibility of the $i$-th subtask. We use the *classification and regression tree* (CART) to infer the precondition function $f_{G_{\mathbf{c}}} : \mathbf{x} \rightarrow \mathbf{e}$ for each subtask based on Gini impurity [Breiman, 1984]. Intuitively, the constructed decision tree is the simplest boolean function approximation for the given input-output pairs $\{\mathbf{x}_t, \mathbf{e}_t\}$. The decision tree is converted to a logic expression (*i.e.*, precondition) in sum-of-product (SOP) form to build the subtask graph.

**Subtask reward inference** To infer the subtask reward $\widehat{G}_{\mathbf{r}}^{\text{MLE}}$ in Equation (5), we model the reward for $i$-th subtask as a Gaussian distribution: $G_{\mathbf{r}}^i \sim \mathcal{N}(\widehat{\mu}^i, \widehat{\sigma}^i)$. Then, the MLE of subtask reward is given as the empirical mean and variance of the rewards received after taking the eligible option $\mathcal{O}^i$ in adaptation phase:

$$\widehat{\mu}_{\text{MLE}}^i = \mathbb{E}\left[r_t|\mathbf{o}_t = \mathcal{O}^i, \mathbf{e}_t^i = 1\right], \tag{6}$$

$$\widehat{\sigma^2}_{\text{MLE}}^i = \mathbb{E}\left[(r_t - \widehat{\mu}_{\text{MLE}}^i)^2|\mathbf{o}_t = \mathcal{O}^i, \mathbf{e}_t^i = 1\right], \tag{7}$$

where $\mathcal{O}^i$ is the option corresponding to the $i$-th subtask. Algorithm 1 outlines the meta-training process.

## 4.3 EVALUATION: GRAPH-REWARD PROPAGATION POLICY

In both meta-training and meta-testing, the agent's adapted behavior is evaluated during the test phase. Following Sohn et al. [2019], we adopted the graph reward propagation (GRProp) policy as an evaluation policy $\pi^{\text{eval}}$ that takes as input the inferred subtask graph $\widehat{G}$ and outputs the subtasks to execute to maximize the return. Intuitively, the GRProp policy approximates a subtask graph to a differentiable form such that we can compute the gradient of return with respect to the completion vector to measure how much each subtask is likely to increase the return. Due to space limitations, we give a detail of the GRProp policy in Appendix. The overall meta-training process is summarized in Appendix.

## 4.4 META-TESTING: MULTI-TASK TASK INFERENCE

**Prior sampling** In meta-testing, MTSGI first chooses the prior task that most resembles the given evaluation task. Specifically, we define the pair-wise similarity between a prior task $\mathcal{M}_G^{\text{prior}}$ and the evaluation task $\mathcal{M}_G$ as follows:

$$\text{sim}\left(\mathcal{M}_G, \mathcal{M}_G^{\text{prior}}\right) = F_\beta\left(\Phi, \Phi^{\text{prior}}\right) + \kappa R\left(\tau^{\text{prior}}\right),$$
(8)

where $F_\beta$ is the F-score with weight parameter $\beta$, $\Phi$ is the subtask set of $\mathcal{M}_G$, $\Phi^{\text{prior}}$ is the subtask set of $\mathcal{M}_G^{\text{prior}}$, $R\left(\tau^{\text{prior}}\right)$ is the agent's empirical performance on the prior task $\mathcal{M}_G^{\text{prior}}$, and $\kappa$ is a scalar-valued weight which we used $\kappa = 1.0$ in experiment. $F_\beta$ measures how many subtasks overlap between current and prior tasks in terms of precision and recall as follows:

$$F_\beta = \left(1 + \beta^2\right) \cdot \frac{\text{precision} \cdot \text{recall}}{(\beta^2 \cdot \text{precision}) + \text{recall}}, \quad (9)$$

$$\text{Precision} = |\Phi \cap \Phi^{\text{prior}}|/|\Phi^{\text{prior}}|, \quad (10)$$

$$\text{Recall} = |\Phi \cap \Phi^{\text{prior}}|/|\Phi|. \quad (11)$$

We used $\beta = 10$ to assign a higher weight to the current task (*i.e.*, recall) than the prior task (*i.e.*, precision).

**Multi-task subtask graph inference** Let $\tau$ be the adaptation trajectory, and $\tau^{\text{p}}$ be the sampled prior adaptation trajectory. Then, we model our evaluation policy as follows:

$$\pi(o|s, \tau, \tau^{\text{p}}) \simeq \pi(o|s, G^\tau)^\alpha \pi(o|s, G^{\text{p}})^{(1-\alpha)}. \quad (12)$$

Due to the limited space, we include the detailed derivation of Equation (12) in Appendix. Finally, we deploy the GRProp policy as a contextual policy:

$$\pi^{\text{eval}}(\cdot|\tau, \tau^{\text{p}}) = \text{GRProp}(\cdot|G^\tau)^\alpha \text{GRProp}(\cdot|G^{\text{p}})^{(1-\alpha)}. \quad (13)$$

Note that Equation (13) is the weighted sum of the logits of two GRProp policies induced by prior $\tau^{\text{p}}$ and current experience $\tau$. We claim that such form of ensemble induces the positive transfer in compositional tasks. Intuitively, ensembling GRProp is taking a union of preconditions since GRProp assigns a positive logit to task-relevant subtask and non-positive logit to other subtasks. As motivated in the Introduction, related tasks often share the task-relevant preconditions; thus, taking the union of task-relevant preconditions is likely to be a positive transfer and improve the generalization. The pseudo-code of the multi-task subtask graph inference process is summarized in Algorithm 2.

## 5 RELATED WORK

**Web navigating RL agent** Previous work introduced MiniWoB [Shi et al., 2017] and MiniWoB++ [Liu et al., 2018] benchmarks that are manually curated sets of simulated toy environments for the web navigation problem. They formulated the problem as acting on a page represented as a Document Object Model (DOM), a hierarchy of objects in the page. The agent is trained with human demonstrations and online episodes in an RL loop. Jia et al. [2019] proposed a graph neural network based DOM encoder and a multi-task formulation of the problem similar to this work. Gur et al. [2018] introduced a manually-designed curriculum learning method and an LSTM based DOM encoder. DOM level representations of web pages pose a significant sim-to-real gap as simulated websites are considerably smaller (100s of nodes) compared to noisy real websites (1000s of nodes). As a result, these models are trained and evaluated on the same simulated environments which are difficult to deploy on real websites. Our work formulates the problem as abstract web navigation on real websites where the objective is to learn a latent subtask dependency graph similar to a sitemap of websites. We propose a multi-task training objective that generalizes from a fixed set of real websites to unseen websites without any demonstration, illustrating an agent capable of navigating real websites for the first time.

**Meta-reinforcement learning** To tackle the few-shot RL problem, researchers have proposed two broad categories of meta-RL approaches: RNN- and gradient-based methods. The RNN-based meta-RL methods [Duan et al., 2016, Wang et al., 2016, Hochreiter et al., 2001] encode the common knowledge of the task into the hidden states and the parameters of the RNN. The gradient-based meta-RL methods [Finn et al., 2017, Nichol et al., 2018, Gupta et al., 2018, Finn et al., 2018, Kim et al., 2018] encode the task embedding in terms of the initial policy parameter for fast adaptation through meta gradient. Existing meta-RL approaches, however, often require a large amount of environment interaction due to the long-horizon nature of the few-shot RL tasks. Our work instead explicitly infers the underlying task parameter

in terms of subtask graph, which can be efficiently inferred using the inductive logic programming (ILP) method and be transferred across different, unseen tasks.

**More Related Works**   Please refer to the Appendix for further discussions about other related works.

# 6   EXPERIMENT

## 6.1   DOMAINS

**Mining**   *Mining* [Sohn et al., 2018] is a 2D grid-world domain inspired by Minecraft game where the agent receives a reward by picking up raw materials in the world or crafting items with raw materials. The subtask dependency in *Mining* domain comes from the crafting recipe implemented in the game. Following Sohn et al. [2018], we used the pre-generated training/testing task splits generated with four different random seeds. Each split set consists of 3200 training tasks and 440 testing tasks for meta-training and meta-testing, respectively. We report the performance averaged over the four task split sets.

**SymWoB**   We implement a symbolic version of the checkout process on the 15 real-world websites such as **Amazon**, **BestBuy**, and **Walmart**, etc.

**Subtask and option policy.** Each actionable web element (*e.g.*, text field, button drop-down list, and hyperlink) is considered as a subtask. We assume the agent has pre-learned the option policies that correctly interact with each element (*e.g.*, click the button or fill out the text field). Thus, the agent should learn a policy over the option.

**Completion and eligibility.** For each subtask, the completion and eligibility are determined based on the status of the corresponding web element. For example, the subtask of a text field is *completed* if the text field is filled with the correct information, and the subtask of a `confirm_credit_info` button is *eligible* if all the required subtasks (*i.e.*, filling out credit card information) on the webpage are completed. Executing an option will complete the corresponding subtask *only if* the subtask is eligible.

**Reward function and episode termination.** The agent may receive a non-zero reward only at the end of the episode (*i.e.*, sparse-reward task). When the episode terminates due to the time budget, the agent may not receive any reward. Otherwise, the following two types of subtasks terminate the episode and give a non-zero reward upon completion:

- **Goal subtask** refers to the button that completes the order (See the green boxes in Figure 1). Completing this subtask gives the agent a +5 reward, and the episode is terminated.
- **Distractor subtask** does not contribute to solving the

given task but terminates the episode with a -1 reward. It models the web elements that lead to external web pages such as `Contact_Us` button in Figure 1.

**Transition dynamics.** The transition dynamics follow the dynamics of the actual website. Each website consists of multiple web pages. The agent may only execute the subtasks that are currently visible (*i.e.*, on the current web page) and can navigate to the next web page only after filling out all the required fields and clicking the continue button. The goal subtask is present in the last web page; thus, the agent must learn to navigate through the web pages to solve the task.

For more details about each task, please refer to Appendix.

## 6.2   AGENTS

We compared the following algorithms in the experiment.

- MTSGI (Ours): our multi-task SGI agent
- MSGI [Sohn et al., 2019]: SGI agent without multi-task learning
- HRL: an Option [Sutton et al., 1999]-based proximal policy optimization (PPO) [Schulman et al., 2017] agent with the gated rectifier unit (GRU)
- Random: a heuristic policy that uniform randomly executes an eligible subtask

More details on the architectures and the hyperparameters can be found in Appendix.

**Meta-training**   In *SymWoB*, for each task chosen for a meta-testing, we randomly sampled $N_{\text{train}}$ tasks among the remaining 14 tasks and used it for meta-training. We used $N_{\text{train}} = 1$ in the experiment (See Figure 8 for the impact of the choice of $N_{\text{train}}$). For example, we meta-trained our MTSGI on **Amazon** and meta-tested on **Expedia**. For *Mining*, we used the train/test task split provided in the environment. The RL agents (*e.g.*, HRL) were individually trained on each test task; the policy was initialized when a new task is sampled and trained during the adaptation phase. All the experiments were repeated with four random seeds, where different training tasks were sampled for different seeds.

## 6.3   RESULT: FEW-SHOT GENERALIZATION PERFORMANCE

Figure 3 and Figure 4 show the few-shot generalization performance of the compared methods on *SymWoB* and *Mining*. In Figure 3, MTSGI achieves more than 75% zero-shot success rate (*i.e.*, success rate at x-axis=0) on all five tasks, which is significantly higher than the zero-shot performance of MSGI. This indicates that the prior learned from the training task significantly improves the subtask graph inference and in turn improves the multi-task evaluation policy. Moreover, our MTSGI can learn a near-optimal policy on all

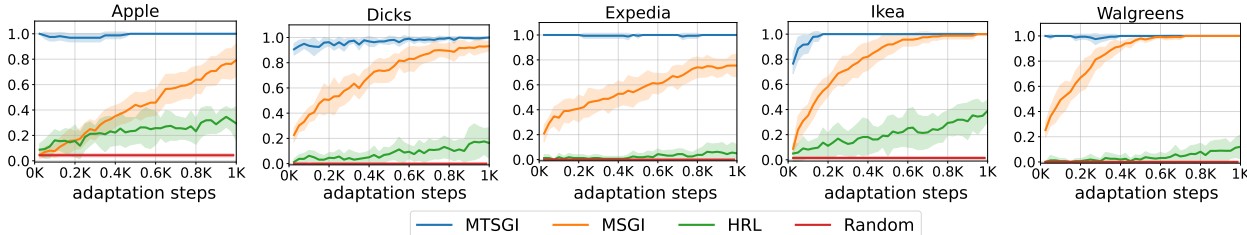

Figure 3: The success rate (y-axis) of the compared methods in the test phase in terms of the environment step during the adaptation phase (x-axis) on *SymWoB* domain. See Appendix for the results on other tasks.

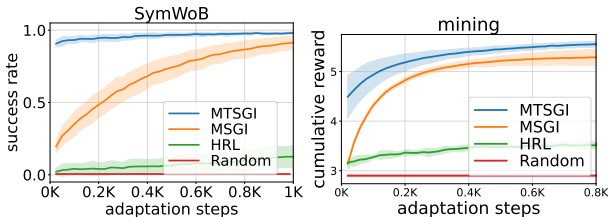

Figure 4: The performance of the compared methods in terms of the adaptation steps averaged over all the tasks in *SymWoB (Left)* and *Mining (Right)* domains.

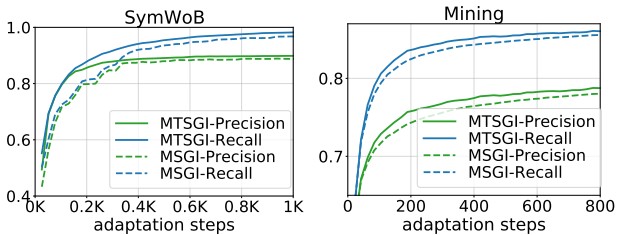

Figure 5: The precision and recall of the subtask graphs inferred by MTSGI and MSGI on *SymWoB* and *Mining*.

the tasks after only 1,000 steps of environment interactions, demonstrating that the proposed multi-task learning scheme enables fast adaptation. Even though the MSGI agent is learning each task from scratch, it still outperforms the HRL and Random agents. This shows that explicitly inferring the underlying task structure and executing the predicted sub-task graph is significantly more effective than learning the policy from the reward signal (*i.e.*, HRL) on complex compositional tasks. Given the pre-learned options, HRL agent can slightly improve the success rate during the adaptation via PPO update. However, training the policy only from the sparse reward requires a large number of interactions especially for the tasks with many distractors (*e.g.*, **Expedia** and **Walgreens**).

### 6.4 ANALYSIS ON THE INFERRED SUBTASK GRAPH

We compare the inferred subtask graph with the ground-truth subtask graph. Figure 6 shows the subtask graph inferred by MTSGI in **Walmart**. We can see that MTSGI can accurately infer the subtask graph; the inferred sub-task graph is missing only two preconditions (shown in

red color) of Click_Continue_Payment subtask. We note that such a small error in the subtask graph has a negligible effect as shown in Figure 3: *i.e.*, MTSGI achieves near-optimal performance on **Walmart** after 1,000 steps of adaptation. Figure 5 measures the precision and recall of the inferred precondition (*i.e.*, the edge of the graph). First, both MTSGI and MSGI achieve high precision and recall after only a few hundred of adaptation. Also, MTSGI outperforms MSGI in the early stage of adaptation. This clearly demonstrates that the MTSGI can perform more accurate task inference due to the prior learned from the training tasks.

### 6.5 ABLATION STUDY: EFFECT OF EXPLORATION STRATEGY

In this section, we investigate the effect of various exploration strategies on the performance of MTSGI. We compared the following three adaptation policies:

- Random: A policy that uniformly randomly executes any eligible subtask.
- UCB: The UCB policy defined in Equation (2) that aims to execute the novel subtask. The exploration parameters are initialized to zero when a new task is sampled.
- MTUCB (Ours): Our multi-task extension of UCB policy. When a new task is sampled, the exploration parameter is initialized with those of the sampled prior.

Figure 7 summarizes the result on *SymWoB* and *Mining* domain, respectively. Using the more sophisticated exploration policy such as MTSGI+UCB or MTSGI+MTUCB improved the performance of MTSGI compared to MTSGI+Random, which was also observed in Sohn et al. [2019]. This is because better exploration helps the adaptation policy collect more data for logic induction by executing more diverse subtasks. In turn, this results in more accurate subtask graph inference and better performance. Also, MTSGI+MTUCB outperforms MTSGI+UCB on both domains. This indicates that transferring the exploration parameters makes the agent's exploration more efficient in meta-testing. Intuitively, the transferred exploration counts inform the agent which subtasks were *under-explored* during meta-training,

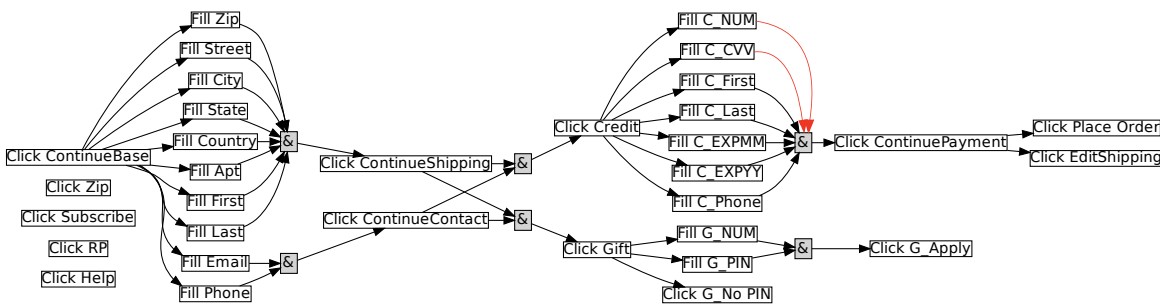

Figure 6: The visualization of the subtask graph inferred by our MTSGI after 1,000 steps of environment interaction on **Walmart** domain. Compared to the ground-truth subtask graph (not available to the agent), there was no error in the nodes and only two missing edges (in red). See Appendix for the progression of the inferred subtask graph with varying adaptation steps.

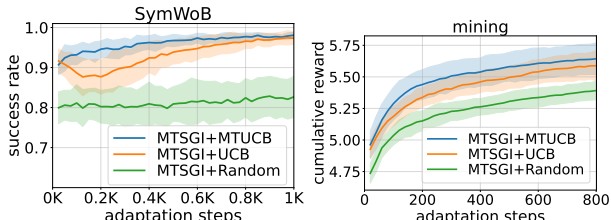

Figure 7: Comparison of different exploration strategies for MTSGI used in adaptation phase for *SymWoB* and *Mining*.

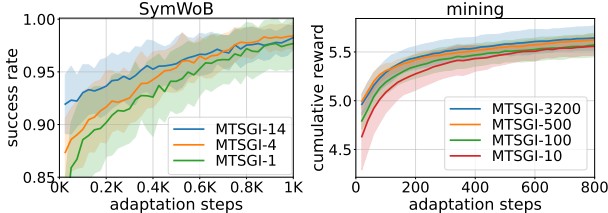

Figure 8: Comparison of different number of priors for MTSGI on *SymWoB* and *Mining*.

such that the agent can focus more on exploring those in meta-testing.

### 6.6 ABLATION STUDY: EFFECT OF THE PRIOR SET SIZE

MTSGI learns the prior from the training tasks. We investigated how many training tasks are required for MTSGI to learn a good prior for transfer learning. Figure 8 compares the performance of MTSGI with the varying number of training tasks: 1, 4, 14 tasks for *SymWoB* and 10, 100, 500, 3200 tasks for *Mining*. The training tasks are randomly subsampled from the entire training set. The result shows that training on a larger number of tasks generally improves the performance. *Mining* generally requires more number of training tasks than *SymWoB* because the agent is required to solve 440 different tasks in *Mining* while *SymWoB* was evaluated on 15 tasks; the agent is required to capture a wider range of task distribution in *Mining* than *SymWoB*. Also, we note that MTSGI can still adapt much more efficiently than all other baseline methods even when only a small number

of training tasks are available (*e.g.*, one task for *SymWoB* and ten tasks for *Mining*).

## 7 CONCLUSION

We introduce a multi-task RL extension of the subtask graph inference framework that can quickly adapt to the unseen tasks by modeling the prior of subtask graph from the training tasks and transferring it to the test tasks. The empirical results demonstrate that our MTSGI achieves strong zero-shot and few-shot generalization performance on 2D grid-world and complex web navigation domains by transferring the common knowledge learned in the training tasks to the unseen ones in terms of subtask graph.

In this work, we have assumed that the subtasks and the corresponding options are pre-learned and that the environment provides a high-level status of each subtask (*e.g.*, whether the web element is filled in with the correct information). In future work, our approach may be extended to a more general setting where the relevant subtask structure is fully learned from (visual) observations, and the corresponding options are autonomously discovered.

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
