# OpenReview forum: "Fast Inference and Transfer of Compositional Task Structures for Few-shot Task Generalization"
_auai.org/UAI/2022/Conference — UAI 2022 Oral_

### Official Review · Reviewer_Dx7q · 2022-04-13

**Q2(1) Originality/Novelty:** 3
**Q2(2) Significance/Impact:** 2
**Q2(3) Correctness/Technical Quality:** 3
**Q2(6) Clarity Of Writing:** 4
**Q6 Overall Score:** 6
**Q8 Confidence In Your Score:** 2

**Q1 Summary And Contributions:**

This paper proposes Multi-Task Subtask Graph Inference (MTSGI) for few-shot learning of tasks that are characterized by a subtask graph (e.g. navigating a website).  MTSGI generalizes a previously learned task structure from training tasks to a new unseen task for faster adaptation and stronger generalization. During meta-training MTSGI models a prior in terms of both an adaptation trajectory and subtask graph, and during meta-testing MTSGI samples the prior that best matches the current task.

**Q2 Assessment Of The Paper:**

More detailed information regarding each of these aspects is given below:

**Q2(4) Quality Of Experiments (Optional):**

3: Good: The experimental evaluation is adequate, and the results convincingly support the main claims.

**Q2(5) Reproducibility:**

3: Good: Key resources (e.g., proofs, code, data) are available and key details (e.g., proofs, experimental setup) are sufficiently well-described for competent researchers to confidently reproduce the main results.

**Q3 Main Strengths:**

* Good empirical results and ablation study
* Clarity of writing

**Q4 Main Weakness:**

* Requirements such as subtask pre-learned options and the environment providing high level status may limit the approaches applicability to different problems.

**Q5 Detailed Comments To The Authors:**

As mentioned in future work, I would like to see MTSGI extended to work in more general settings and have subtask options be discovered or learned.

**Q7 Justification For Your Score:**

I'm don't see anything wrong with this paper and so recommend acceptance, but question it's significance/impact given the need for pre-learned subtask options and requirement that the environment provide high level status which is why I went with weak accept.  I'm also not that sure about the novelty of the paper which is partly why I have a lower confidence score.

**Q9 Complying With Reviewing Instructions:**

1: Yes.

---

### Official Review · Reviewer_wWig · 2022-04-14

**Q2(1) Originality/Novelty:** 3
**Q2(2) Significance/Impact:** 2
**Q2(3) Correctness/Technical Quality:** 3
**Q2(6) Clarity Of Writing:** 3
**Q6 Overall Score:** 7
**Q8 Confidence In Your Score:** 2

**Q1 Summary And Contributions:**

The paper proposes multi-task subtask graph inferences (MTSGI), a meta reinforcement learning approach aiming to improve inference on the testing task, by constructing a prior on the training task. The goal is to tackle the scenario where unseen testing tasks are present, transferring the knowledge from training. The approach relies on modeling a task by a series of subtasks. Experiments are evaluated on real-world complex structure data.

**Q2 Assessment Of The Paper:**

More detailed information regarding each of these aspects is given below:

**Q2(4) Quality Of Experiments (Optional):**

3: Good: The experimental evaluation is adequate, and the results convincingly support the main claims.

**Q2(5) Reproducibility:**

3: Good: Key resources (e.g., proofs, code, data) are available and key details (e.g., proofs, experimental setup) are sufficiently well-described for competent researchers to confidently reproduce the main results.

**Q3 Main Strengths:**

The paper is clearly written and structured. The methodology is well presented, providing the necessary background, as well as the experiments. The experiments are properly designed and demonstrate the validity of the proposed approach. The paper is technically sound and although built on previous approaches, there is consistent novelty in the framework.

**Q4 Main Weakness:**

Please, see Q5.

**Q5 Detailed Comments To The Authors:**

There are some considerations, mostly in terms of clarification, which could improve the quality of the paper.

1) In equation 8, can the author elaborate more on the choice of this function? Are there alternative options?
2) I would recommend to better highlight the difference with previous work, especially with MSGI. Including, clearly stating the main contributions of the paper, as well as the novel aspects
3) Regarding the ablation study on the number of training tasks required, I would recommend to compare the results with MSGI.
4) Could the authors comment on the selection/properties of the test tasks. For example, is there any consideration or evaluation with respect to their similarity with training?

**Q7 Justification For Your Score:**

Overall, I believe the paper is technically solid, while the exposition and organization is clear. The experiments are also well structured and support the theoretical claim.

**Q9 Complying With Reviewing Instructions:**

1: Yes.

---

### Official Review · Reviewer_GzLr · 2022-04-16

**Q2(1) Originality/Novelty:** 3
**Q2(2) Significance/Impact:** 3
**Q2(3) Correctness/Technical Quality:** 3
**Q2(6) Clarity Of Writing:** 3
**Q6 Overall Score:** 7
**Q8 Confidence In Your Score:** 3

**Q1 Summary And Contributions:**

The authors provide a multi-task subtask graph inference (MTSGI), which first infers the common high-level task structure in terms of the subtask graph in meta-training, and uses it as a prior to improve the task inference in meta-testing. Compared with the previous meta-RL methods trying to directly infer the unstructured task embedding, this framework improves few-shot generalization performance.

**Q2 Assessment Of The Paper:**

More detailed information regarding each of these aspects is given below:

**Q2(4) Quality Of Experiments (Optional):**

2: Fair: The experimental evaluation is weak: important baselines are missing, or the results do not adequately support the main claims.

**Q2(5) Reproducibility:**

2: Fair: Key resources (e.g., proofs, code, data) are unavailable but key details (e.g., proof sketches, experimental setup) are sufficiently well-described for an expert to confidently reproduce the main results.

**Q3 Main Strengths:**

1. The framework of MTSGI is logical and natural. This work focus on solving and generalizing unseen compositional sparse-reward tasks with complex subtask dependencies without human supervision. Effectively, based on the prior, the accuracy of this meta-RL model is significantly improved under this framework.
2. This work efficiently infers the underlying task parameter in terms of subtask graph using the inductive logic programming (ILP) method and be transferred across different, unseen tasks. It avoids a large number of environmental interactions and speeds up the adaptation of parameters in few-shot Learning.
3. The authors have a clear definition of the research task and the experiments compare the performance of your model with a representative baseline method, which makes the results informative.

**Q4 Main Weakness:**

1. The experimental details and result analysis are inadequate. It is not advisable to include some important experimental details in the appendix. And it would be perfect if you comprehensively analyze the experimental results horizontally and vertically.

2. It would be better to give more details about precondition inference, such as the procedure of inductive logic programming and classification and regression tree.

**Q5 Detailed Comments To The Authors:**

Taking SymWoB as an example, the authors mention that for each task chosen for a meta-testing, you randomly sampled a task among the remaining 19 tasks and used it for meta-training. Then for the different tasks selected for training, there may be different effects on the subtask graph inference results of the current test task. How this is dealt with in the experiments?

**Q7 Justification For Your Score:**

The proposed method is novel and the writing of paper is clear.

**Q9 Complying With Reviewing Instructions:**

1: Yes.

---

### Decision · Program_Chairs · 2022-05-15

**Decision:**

Accept (Oral)

**Comment:**

Meta Review: The authors propose a multi-task subtask graph inference (MTSGI) for few-shot learning of tasks, which first infers the common high-level task structure in terms of the subtask graph in meta-training. Compared with the previous meta-RL methods, this proposed model improves few-shot generalization performance. Experiments are sufficient and evaluated on real-world complex structure datasets.